# Carcasses at Fixed Locations Host a Higher Diversity of Necrophilous Beetles

**DOI:** 10.3390/insects12050412

**Published:** 2021-05-04

**Authors:** Christian von Hoermann, Tomáš Lackner, David Sommer, Marco Heurich, M. Eric Benbow, Jörg Müller

**Affiliations:** 1Department of Conservation and Research, Bavarian Forest National Park, Freyunger Str. 2, 94481 Grafenau, Germany; Tomas.Lackner@npv-bw.bayern.de (T.L.); Joerg.Mueller@npv-bw.bayern.de (J.M.); 2Department of Zoology, Faculty of Science, Charles University, Vinicna 7, 12844 Praha, Czech Republic; dejv.sommer@gmail.com; 3Department of Ecology, Faculty of Environmental Sciences, Czech University of Life Sciences in Prague, Kamycka 1176, 16521 Praha, Czech Republic; 4Department of Visitor Management and National Park Monitoring, Bavarian Forest National Park, Freyunger Str. 2, 94481 Grafenau, Germany; Marco.Heurich@npv-bw.bayern.de; 5Chair of Wildlife Ecology and Management, Albert-Ludwigs University of Freiburg, Tennenbacher Str. 4, 79106 Freiburg, Germany; 6Institute for Forest and Wildlife Management, Inland Norway University of Applied Science, NO-2480 Koppang, Norway; 7Department of Entomology, Department of Osteopathic Specialties, AgBioResearch and Ecology, Evolution and Behavior Program, Michigan State University, East Lansing, MI 48824, USA; benbow@msu.edu

**Keywords:** carrion, Coleoptera, decomposition, forest, indicator species, necrobiome, scavenger, succession, trapping

## Abstract

**Simple Summary:**

Whereas vertebrate scavengers have a higher diversity reported at randomly placed carcasses, the drivers of insect diversity on carrion, such as the exposure type (fixed versus random) or the carrion species, are still incompletely understood. We analyzed beetle diversity at differently exposed carcasses in the low-range mountain forest of the Bavarian Forest National Park in Germany. We tested if scavenging beetles, similarly to vertebrate scavengers, show a higher diversity at randomly placed carcasses compared to easily manageable fixed places. Ninety-two beetle species at 29 exposed wildlife carcasses (roe, red deer, and red foxes) were detected. Beetle diversity was higher at fixed locations possessing extended highly nutrient-rich cadaver decomposition islands as important refuges for threatened red-listed species, such as *Necrobia violacea* (Coleoptera: Cleridae). Particularly noticeable in our insect traps were the following two rare species, the “primitive” carrion beetle *Necrophilus subterraneus* (Coleoptera: Agyrtidae) and the false clown beetle *Sphaerites glabratus* (Coleoptera: Sphaeritidae). In Europe, only the species *S. glabratus* out of the genus Sphaerites is present. This emphasizes the importance of carrion for biodiversity conservation. We clearly show the relevance of leaving and additional providing wildlife carcasses in a dedicated place in protected forests for preserving very rare and threatened beetle species as essential members of the decomposing community.

**Abstract:**

In contrast to other necromass, such as leaves, deadwood, or dung, the drivers of insect biodiversity on carcasses are still incompletely understood. For vertebrate scavengers, a richer community was shown for randomly placed carcasses, due to lower competition. Here we tested if scavenging beetles similarly show a higher diversity at randomly placed carcasses compared to easily manageable fixed places. We sampled 12,879 individuals and 92 species of scavenging beetles attracted to 17 randomly and 12 at fixed places exposed and decomposing carcasses of red deer, roe deer, and red foxes compared to control sites in a low range mountain forest. We used rarefaction-extrapolation curves along the Hill-series to weight diversity from rare to dominant species and indicator species analysis to identify differences between placement types, the decay stage, and carrion species. Beetle diversity decreased from fixed to random locations, becoming increasingly pronounced with weighting of dominant species. In addition, we found only two indicator species for exposure location type, both representative of fixed placement locations and both red listed species, namely *Omosita depressa* and *Necrobia violacea*. Furthermore, we identified three indicator species of Staphylinidae (*Philonthus marginatus* and *Oxytelus laqueatus*) and Scarabaeidae (*Melinopterus prodromus*) for larger carrion and one geotrupid species *Anoplotrupes stercorosus* for advanced decomposition stages. Our study shows that necrophilous insect diversity patterns on carcasses over decomposition follow different mechanisms than those of vertebrate scavengers with permanently established carrion islands as important habitats for a diverse and threatened insect fauna.

## 1. Introduction

Animal carrion is the most nutrient-rich form of dead organic matter and decomposes at a fast rate [1,2,3]. These two key qualities of high nutrient concentration and accelerated temporal dynamics make carrion a highly important component of the detritus pool [4]. To obtain a broader recognition of the similarities and differences of decomposing organic matter of any type, Benbow [5] used the term necromass as synonymous with the holistic definition of detritus by Moore [2]. Necromass of any size, shape, and quality (e.g., different diameters and types of deadwood or different carrion types, such as foxes, beavers, red deer, reptiles, or amphibians) dictates many food webs and has a significant bottom-up importance for ecosystem functioning [6,7,8]. Additionally, necromass forms a resource aggregate for behaviors and interactions among members of the necrobiome, including species of microbes, invertebrates, and vertebrate scavengers [5]. Such complex communities shape whole ecosystems across spatial and temporal scales [9].

However, the drivers of communities in different taxa and how they use and occupy different forms of necromass are still incompletely understood; this is particularly true for carrion in contrast to deadwood (e.g., see References [10,11]). Insects are the most species-rich and abundant animals found on carrion [12]. They deliver a critical ecosystem service by consuming, dispersing, and recycling carrion nutrients [3]. When insects are absent, the rate of decomposition can be significantly delayed [3,13] and can affect microbial processing of carrion [14,15]. Consequently, for continuous functioning of ecosystem processes and services, it is of great importance to preserve the biodiversity of the carcass-associated insect fauna and therefore to identify its drivers.

For any species, non-consumptive mortality, such as natural senescence or disease-related death, is a natural part of its population dynamics and contributes to the detritus pool and its associated nutrient and energy dynamics [7]. Besides spatially and temporally random distributed death events, the input of vertebrate carrion can also result in spatiotemporally concentrated mass die-off effects. For instance, thousands of kilograms of moose (*Alces alces*) carcasses per square kilometer in Michigan, USA [16], or millions of kilograms of elephant (*Loxodonta africana*) cadavers during a mass starvation in Kenya [17] have been reported [18]. Increases in disease, including epidemics, could produce such mass die-off events with disruption effects on the spatial and temporal availability of carrion within the ecosystem [8,18]. Understanding the spatial availability of both individual carcasses and mass mortality events is becoming increasingly recognized to have ecosystem function and resource management relevance [18].

Wilmers [19] and Cortés-Avizanda [20] showed that temporal aggregation of carcasses reduced the diversity and evenness of carrion consumption among vertebrate scavengers. However, additional studies are needed to examine and directly manipulate the effects of carrion exposition type (randomly versus predictable fixed locations, e.g., Reference [21]) on invertebrate and vertebrate scavenger community structure. For instance, carrion exposition type was reported as an important driver of avian scavenger diversity [20], where it was shown that randomly placed, unpredictably located cadavers attracted a more diverse (Shannon index) avian scavenger community. The authors suggested that randomly exposed ungulate cadavers (unpredictable trophic pulses) served as a precondition for the prevention of interspecific competition with a positive effect on scavenger biodiversity [20]. They also suggested that facilitatory processes (“occurrence of positive interspecific relationships within the guild”) were responsible for the high detected diversity patterns at randomly exposed carrion through species coexistence [20,22]. In predictable conditions, the dominant specialist species can arrive earlier and is able to monopolize the carrion resource [20]. Fitting with this result, Wilmers [19] reported for the Yellowstone National Park a higher vertebrate scavenger diversity at randomly appearing carrion remains caused by wolf kills compared with predictable remains at fixed locations that human hunters established. The reintroduction of wolves in the Yellowstone National Park in 1995 increased the abundance of 13 vertebrate scavenger species by the regular offer of wolf kills [19,23]. Wilmers [19] found a dominance of highly competitive species, such coyotes (*Canis latrans*) at the spatially and temporally more dispersed wolf kills compared to the spatiotemporally highly aggregated and predictable carrion remains caused by human hunting that were dominated by species with large feeding radii such as ravens and the bald eagle. Given the findings that carcass exposure location and predictability affect vertebrate scavenger communities a central question still exists about if and how invertebrate necrophilous communities may also respond to the spatial availability of vertebrate carrion with potential consequences for the competition between scavengers and necrophagous insects.

For necrophilous invertebrates it has been widely documented that there is a predictable community assembly and succession through decomposition [4,24,25,26,27,28,29]. This temporal succession theory [24] has a focus on explaining species temporal occurrences [4]. For carrion insects, the succession can be described as following: during the first stage of decomposition, the “fresh stage”, the first arriving insects are mostly Calliphoridae and Sarcophagidae flies [30,31]. Their eggs and larvae need moist tissue for successful development [32]. During the next stage, the “bloated stage” (inflated abdomen through gaseous by-products of putrefaction), significant calliphorid maggot masses can be observed [30,31]. In the “post-bloating stage” (=active decay; skin rupture and release of trapped putrefactive gases), not only large feeding masses of fly maggots, but also predatory beetles of the Silphidae, Staphylinidae, and Histeridae families can be observed foraging on fly larvae. At the end of this stage, most of the maggots have left carrion for pupation [30,31]. Late “post-bloating” is preferred by adult dermestid beetles, feeding on the remaining skin and ligamentous tissue [12,33]. In the next two stages, the “advanced decay stage” (most of the flesh has disappeared, and some soft tissue remains in the abdomen) and the “dry remains stage” (only bones, hair, and remains of dried-out skin remain), coleopteran taxa such as Cleridae, Dermestidae, Scarabaeidae, and Trogidae dominate the fauna of the cadaver [31,34,35,36]. Thus, the community of invertebrates that are attracted to, use and colonize carrion is defined by a changing resource as it is rapidly recycled in the ecosystem. In this study, we hypothesized that the type (i.e., vertebrate species), position, and predictability of these resources in a forest ecosystem will shape the necrophagous insect communities that occupy and use it during decomposition. We predicted that randomly placed carcasses would attract and reflect a more diverse insect community compared to predictable carcass exposure locations, similar to that reported for vertebrates. We further investigated the impact of the decomposition stage and carcass biomass on diversity, as well as all on characteristic species.

## 2. Materials and Methods

### 2.1. Study Site Description

Cadaver exposition and associated insect trapping were conducted throughout the Bavarian Forest National Park (BFNP) in Eastern Bavaria, Germany. Together with the *Šumava* National Park, Czech Republic, the BFNP forms the Bohemian Forest Ecosystem, one of the largest strictly protected woodlands of Central Europe. A humid and cold continental climate is prevailing with some maritime influence from the West. The annual air temperature ranges between 3.9 and 8.6 °C. The annual precipitation varies from 400 mm up to 2500 mm and elevation ranges from 370 m above sea level (a.s.l.) in the valleys up to 1456 m a.s.l. along the mountain ridges. From October to May, a permanent snow cover lasts up to 7 months on the mountain peaks and up to 5 months from November to April in the valleys. The forested part of the BFNP amounts to 98% and 2% are open raised bogs, water bodies, abandoned mountain pastures, or boulder fields [37]. The forested area is mostly covered by mixed mountain forest, consisting of Norway spruce (*Picea abies*), Silver fir (*Abies alba*) and European beech (*Fagus sylvatica*), and mountain spruce forest. In high montane elevation, above the grass and often spruce-rich mixed forests, are Hercynian mountain forests (*Piceion*) dominated by Norway spruce and accompanied by sycamore (*Acer pseudoplatanus*) and mountain ash (*Sorbus aucuparia*) [38]. Through cold air flowing down the mountain slopes and accumulating in damp valley basins, ground frosts are possible in these zones even in summer months. The large mammal species pool (as an omnipresent source of carrion) includes the herbivores, red deer (*Cervus elaphus*) and roe deer (*Capreolus capreolus*); the omnivore wild boar (*Sus scrofa*); and the carnivores, Eurasian lynx (*Lynx lynx*) and the gray wolf (*Canis lupus*). The most abundant carcass-visiting mesopredators are the red fox (*Vulpes vulpes*) and the pine marten (*Martes martes*) [21]. As part of the invertebrate species pool, more than 2000 beetle species were documented [39].

### 2.2. Experimental design

#### 2.2.1. Wildlife Carcass Exposure

Carcasses of 6 red deer, 18 roe deer, and 5 foxes were placed in the BFNP over the months of June to November 2018 at four different fixed locations (a total of 12 carcasses) and 17 different random sites (a total of 17 carcasses) as follows (Appendix A Appendix A). On the first Tuesday of every month, two deer carcasses were placed on two of the four fixed locations and 2 up to 4 carcasses (2 deer and in addition (if available) 1 or 2 foxes; Appendix A Appendix A) were placed once on each of the overall 17 random sites. In each subsequent month, the other two fixed locations were fitted with carrion in an alternating way. Carcass numbers and exposed species per month differed because of irregularities in carcass supply. Roe deer and fox cadavers were obtained from wildlife–vehicle collisions that randomly took place at National Park roads, whereas red deer carcasses originated from wildlife control measures in the National Park [40]. The carcasses ranged from freshly killed to previously frozen at −20 °C and were all defrosted for four days to represent the fresh stage of decomposition at the time of exposition (day 0). Carrion weight ranged from 4.7 (single fox carcass) to 110 (single red deer carcass) kilograms (Appendix A Appendix A). One hind limb of each deer and fox carcass was fixed with a cable tied to a wooden stick. After total carcass depletion, bones and remains of furs stayed on site. We mounted data loggers (Thermochron iButton, Whitewater, WI, USA) at each carrion exposition site to record the temperature and humidity of the carrion microhabitat every 30 min during the whole exposition period of 30 days. In parallel to the carrion exposition, a total of 16 control sites (4 activated per month; Appendix A Appendix A) comprising 12 control sites to the corresponding random locations and four control sites of the same type (differing only in location) to the corresponding fixed locations were installed. On each single control site without a carcass, two pitfall traps (see next subheading for more details) and one camera trap were installed. Controls were needed to capture the prevailing and not necessarily carcass-associated insect and vertebrate fauna of the habitat. All carrion exposition and control sites were arranged in half-open mixed montane forest stands and were sufficiently spaced at a minimum distance of 1000 m to avoid cross interactions among individual cadavers. A minimum distance between sampling plots of 200 m is already sufficient to achieve spatial independency in assemblages of flying insects [41,42]. Site elevation ranged from 643 to 1126 m above sea level (Appendix A Appendix A).

#### 2.2.2. Pitfall Trap Installation and Beetle Sampling

Two pitfall traps for trapping of necrophilous insects were installed directly at each wildlife carcass. One pitfall trap was mounted adjacent to the mouth of the carcass, with the other one being adjacent to its anus. Consequently, two important settlement areas (head and anus) for cadaver-inhabiting insects were covered [43]. For pitfall trapping, two ground-level plastic cups were stacked inside each other (half-liter PLA cups; diameter, 95 mm; height, 151.2 mm; Huhtamaki Foodservice GmbH, Alf/Mosel, Germany). For the reduction of surface tension, the inner cup was filled with an odorless soapy solution (one droplet of detergent, Klar EcoSensitive, AlmaWin, Winterbach, Germany). Each single trap was equipped with a rain cover (constructed at BFNP) to avoid overflow. The same procedure was used at the control sites without carcasses. For comparison purposes, the distance of the two control pitfall traps at one single capture site corresponded to the distance between head and anus of the exposed respective wildlife species. A total of 7 trap content collection events per exposed carcass and control were conducted over the decomposition period at 2, 4, 6, 9, 16, 23, and 30 days after day 0 of exposure [36,44]. All distinct decomposition stages, based on large-scale succession data in the literature [28,36,44,45], were covered by these sampling intervals. To guarantee a constant sample period for each trapping event, the pitfall traps were opened by removing the lid (PLA dome-covers for smoothie cups; diameter, 95 mm; Huhtamaki Foodservice GmbH, Alf/Mosel, Germany) at 48 h before content collection. Consequently, each beetle sampling event lasted 48 h. All exposed wildlife cadavers were photo-documented at each trap-emptying event for later morphological assessment and classification of decomposition stages [33].

Immediately after each trap-emptying event, we transferred all collected insect individuals into 70% ethanol (VWR International GmbH, Darmstadt, Germany) for later presorting into the following beetle groups: Silphidae, Staphylinidae, Scarabaeidae, Histeridae, Dermestidae, and “Coleoptera rest” (all other beetle taxonomic groups) and subsequent species identification by external specialists. From these families, all individuals were identified to species level and stored at the BFNP. For any single trapping event, all data for the two traps were pooled (no separate consideration of mouth and anus traps in later analyses). The same procedure was conducted for the control sites. The sampling campaign resulted in a total of 343 sample units (203 carrion sample units and 140 control sample units). All 343 samples from 37 plots formed the basis for subsequent statistical analyses.

### 2.3. Statistics

All analyses were conducted in R and R Studio versions 4.0.3 [46]. To test the effects of carcass exposure location (randomly and fixed location) and controls on overall abundance of obligate, facultative necrophilous, and guest beetles, Poisson-GLM models (including multcomp for post hoc multiple comparison testing) were applied.

To compare species diversity between the exposition types (random versus fixed location) with respect to early, intermediate, and late decay successional stages, we used an individual based rarefaction–extrapolation approach to estimate necrophilous beetle diversity accounting for sampling effort. Hill [47] unified different diversity indices in the Hill numbers, using an increasing weighting from rare to dominant species. We used this approach and compared diversities for three values of q (q = 0, species richness; q = 1, exponential of Shannon’s entropy index; q = 2, inverse of Simpson’s concentration index). Computations were implemented with the iNEXT package [48,49]. Finally, we applied indicator species analysis (*p* < 0.05) to identify obligate and facultative necrophilous species characteristics for both carrion placements (random versus fixed location) and with respect to early, intermediate, and late decay successional stages and carrion species (small, intermediate, and high carrion biomass), using the multipatt function in the add-on package indicspecies [50].

## 3. Results

In total, we identified 92 beetle species from 12,879 individuals on 29 exposed wildlife cadavers (Table 1 and Figure 1), compared to control locations that had only 23 species and 92 beetle individuals (Table 1).

There was a noticeable variation in rank abundance, numbers of individuals, and beetle species on carrion between the exposition sites (Figure 2).

Rank–abundance distribution at fixed locations was more balanced (more flattened rank abundance curve) compared to random sites (steeper curve progression), showing a higher species richness and no striking abundance of single species at fixed locations (Figure 2). Insect numbers ranged from 0 to 1213, collected from one single random location with a 7 kg fox cadaver in November 2018 and a 19.1 kg roe deer cadaver site (fixed location) in June 2018, respectively. The number of necrophilous beetle species per site varied from 0 at a single random site with the same 7 kg fox cadaver in November 2018 to 84 species at a random site with a 21 kg roe deer cadaver in August 2018.

Carcass at both random and fixed locations attracted significantly more necrophilous beetles than control sites (Poisson-GLMM with Tukey contrasts (*p* < 0.05): Randomly versus Fixed location, *z* = 0.684, *p* = 0.773; Randomly versus Control, *z* = −7.326, *p* < 0.001; Fixed location versus Control, *z* = −7.386, *p* < 0.001; Figure 1). Rarefaction–extrapolation analyses revealed higher beetle diversity at fixed than random places, increasingly pronounced with increasing weighting of dominant species along the three Hill numbers (well-separated confidence intervals for Simpson diversity (q = 2); Figure 3 and Appendix A Appendix A).

We detected the highest diversity of common (q = 1) and dominant (q = 2) beetle species during middle decay stages (bloated, active decay) at fixed locations (Figure 3 and Appendix A Appendix A), but the lowest diversity was in middle decay stage carcasses at random sites, which was valid for Hill-numbers 1 (common beetle species) and 2 (dominant beetle species) (Figure 3 and Appendix A Appendix A). For rare species, the highest diversity was during fresh carcass decay stage at fixed locations, whereas the lowest diversity was found during fresh stage at random sites (Figure 3, q = 0). However, confidence intervals of the rarefaction and extrapolation curves highly overlapped for species richness (q = 0; Figure 3 and Appendix A Appendix A), indicating a similar estimated diversity of rare beetle species across treatment types and decay stages.

Indicator species analysis revealed two indicator species for exposition site, both for fixed locations (Appendix A Appendix A) and both were red listed species *Omosita depressa* (IndVal = 0.662; *p* = 0.004) and *Necrobia violacea* (IndVal = 0.540; *p* = 0.004) [51].

Two indicator species of Staphylinidae (*Philonthus marginatus*; IndVal = 0.629; *p* = 0.011 and *Oxytelus laqueatus*; IndVal = 0.568; *p* = 0.017) were detected for high carcass mass (heavy red deer carrion) at random sites (Figure 4 and Appendix A Appendix A) and one indicator species of Scarabaeidae (*Melinopterus prodromus*; IndVal = 0.545; *p* = 0.012) at fixed locations (Figure 4 and Appendix A Appendix A).

One geotrupid species (*Anoplotrupes stercorosus*; IndVal = 0.866; p = 0.017) was an indicator species for intermediate (bloated and active decay) and advanced decay and dry stages during carrion decomposition at random sites (Figure 4 and Appendix A Appendix A).

## 4. Discussion

We found significant higher necrophilous beetle species diversity at fixed place carcasses, most pronounced with dominant species. Our finding is the opposite reported for vertebrate scavengers at carrion resources [19,20] and thus does not support our vertebrate driven hypothesis of a more diverse insect community at random placed carcasses.

VanLaerhoven [52] stated that increases in abundance or carrion amount (size of carrion resources) should increase the number of trophic levels within local food webs, similar to the carrion associated species diversity. For necrophilous coleopteran taxa, our result of a higher diversity at fixed locations with constantly high carrion placement rates (increasing carrion abundance and mass over time) matches with the statement of VanLaerhoven [52]. In contrast to vertebrate scavenger communities, where the dominant species, the Griffon vulture (*Gyps fulvus*), arrives earlier and is able to monopolize the carrion resource in predictable conditions, such as fixed locations [20], we assumed that competition with negative effects on visitor diversity at fixed locations would have a minor, negligible or inverse relationship with necrophilous beetle abundance and diversity. Our results show the highest diversity in dominant beetle species during carcass bloated and active decay at fixed locations. Silphidae are part of that dominant species pool (Table 1, Figure 2), are effective predators of fly larvae [53,54,55,56], and were well established in high numbers at fixed locations (as part of the local background species pool). There increased abundance can reduce the immense maggot masses on carrion during bloated and active decay (maggot mass dominated stages [31]) by direct consumption. In turn, such direct predatory interactions could structure overall species diversity and carrion community dynamics [52,57]. Moreover, the presence of members of other insect taxa, such as ants and wasps (both Hymenoptera), has frequently been reported to delay colonization of flies and change succession and colonization patterns [58,59,60]. We suggest that a quantitative reduction of substrate degrading maggot masses releases niche space for a more diverse necrophilous beetle community. This also applies to rapid carrion monopolization by large vertebrate scavengers, such as lynx or vultures, leading to soft-tissue removal and thus preventing extended blowfly activity [61,62]. Facilitation (or commensalism) was originally defined by Connell [63] as early colonists (e.g., silphids or avian scavengers are immediately available on site at fixed locations), making the environment (e.g., carcass resource) more suitable for later colonists and is one potential mechanism for succession. These early colonists have been called ecosystem engineers [64].

Regarding carrion volume, Moleón [65] suggested that increased carrion biomass resulting from vertebrate scavenger reductions at carnivore carcasses enables a successional insect community of the remains. This was confirmed by a carnivore carcass avoidance-study of Muñoz-Lozano [66], showing a well-structured and diverse insect community consisting of necrophages, omnivores, and necrophilous predators and parasitoids. In contrast to the studies of Moleón [65] and Muñoz-Lozano [66], no carnivore carcasses were exposed at our fixed locations, and access by vertebrate scavengers, such as red foxes, pine martens, or Eurasian lynx, to the exposed roe and red deer carcasses took place [21]. However, the bait site situation in the BFNP with regular and year-round carrion supply guarantees a localized increased carrion volume over time despite the vertebrate scavenger visitors. Consequently, besides the above-discussed facilitation hypotheses, a highly diverse successional insect community could be a result of the total carrion volume, commensurate with the results of Moleón [65] and Muñoz-Lozano [66].

Carrion exposition type (random versus fixed), cadaver species (offer of carrion biomass), and decomposition stages of wildlife carcasses revealed characteristic indicator species. We detected two red-listed [51] indicator species for fixed locations, namely the nitidulid species *Omosita depressa* and the clerid species *Necrobia violacea*. *Necrobia violacea* is a carrion associated saprophagous and predaceous beetle [67]. It feeds on dermestid larvae [68] and is characteristic for the long-lasting existence of advanced decay and dry remains [69,70,71] and therewith associated cadaver decomposition islands (CDIs [72]) as a typical appearance at fixed locations with high carrion turnover rates. Consequently, we suggest the extended highly nutrient-rich (pronounced increase in soil nitrogen concentration and other nutrients, such as calcium, magnesium, and potassium [72]) CDIs at fixed locations provide important refuges for threatened red-listed species, such as *N. violacea*. This is substantiated by the small catching numbers of only one individual at overall 17 random sites compared to 44 specimens at a total of four fixed locations during our trapping campaign in the BFNP, emphasizing the importance of the bait site situation for this beetle species. Haelewaters [71] reported Cleridae as the least abundant during their exposition study of five car-killed roe deer, which are typical random carcasses. Only one specimen was collected during the dry-remains stage [71]. In the Czech Republic, Kočárek [69] collected *N. violacea* as the sole Cleridae species at exposed rat carcasses. Both examples, together with our results, emphasize the importance of fixed feeding sites when considering carrion subsidies as a measure for preserving clerid diversity in a forest ecosystem.

The Nitidulidae family of our second detected red-list indicator species for fixed locations, *O. depressa*, was reported by Benbow [73] as an indicator taxon for active decay of swine carcasses. This supports our assumption about the importance of a well-established and long-lasting CDI situation for beetle diversity being typical for fixed locations. Furthermore, Saloña [74] reported *O. depressa* for the first time in the Iberian Peninsula as part of the edaphic fauna after a corpse was removed. They stated that soil-related fauna remaining for days after the removal of a corpse (as very typical for fixed exposition sites; von Hoermann [75], personal observation in the BFNP) could possibly deliver information related to death circumstances during crime scene investigations. In the literature, *O. depressa* is described as a species living on bones and in dry carcasses [76], both of which are typical long-lasting remains one can encounter at regularly carrion-loaded fixed locations. *Omosita depressa* was also described as very local and scarce sap feeding species in a heavily wooded area in Scotland containing significant amounts of dead wood [77].

In line with our results of two staphylinid species and one Scarabaeidae species as indicative for high carrion biomass, Dekeirsschieter [78] identified *O. laqueatus* on high carrion biomass (large domestic pig carcasses) during spring in a temperate forest biotope. This underlines the importance of cadaver size for specific necrophilous species of the carrion entomofauna. Interestingly, Vindstad [79] found *O. laqueatus* in large deadwood biomass as well. After a devastating outbreak of the moth *Epirrita autumnata* (Lepidoptera: Geometridae) in North Norway, Vindstad [79] identified *O. laqueatus* as not saproxylic beetle species inside of the vast quantities of deadwood with a significant 7.5-fold higher abundance compared to not affected forest at the periphery of the moth outbreak. The findings of Vindstad [79] and Deikeirsschieter [78] and the result of our study show *O. laqueatus* as an indicator species for both large deadwood and carrion biomass (red deer and domestic pig carcasses). *Oxytelus laqueatus* has been recorded in many types of decaying organic matter, even from sap runs [80].

Large maggot masses at large vertebrate carrion (=high cadaver biomass) are preferred feeding sites for necrophilous and predatory staphylinid beetles [31]. Most carrion-associated staphylinid beetles are predators that feed on eggs and larvae of Diptera at active decay and advanced decay stages or comprise saprophagous species feeding on cadaveric fluids and parasitizing dipteran pupae, such as *Aleochara curtula* [18,81,82]. The second staphylinid species we detected as an indicator of high cadaver biomass (carcasses of the large ungulate red deer), *P. marginatus*, is predatory and polyphagous. It feeds on dipteran eggs, larvae, and even adults [83]. Weithmann [84] found the highest staphylinid beetle abundance in one of the German red deer areas, the Schorfheide-Chorin Biosphere Reserve in Brandenburg. The authors concluded that higher numbers of large vertebrates, such as red deer, may increase the occurrence of large carrion in these forests, generating large dipteran larvae masses as prey for the large-sized and predatory *Philonthus* species [84]. Accordingly, we detected the *Philonthus* species *P. marginatus* as an indicator species for red deer carrion in the red deer area BFNP in Southeast Germany. This emphasizes the importance of high cadaver biomass subsidies (e.g., large ungulate carcasses) for establishing a diverse and highly abundant staphylinid community over time, as confirmed for the vertebrate scavenger diversity as well [21].

As indicative for intermediate (bloated and active decay) and late (advanced decay and dry remains) carrion decomposition stages we detected the geotrupids dung beetle species *Anoplotrupes stercorosus*. *Anoplotrupes stercorosus* is a very frequent copronecrophagous forest species and can be encountered in all kinds of dung, on carrion and in old fungi [45,85,86,87]. Our results concur with those of von Hoermann [36], who detected highest abundance of necrophilous dung beetles in advanced decay stages of exposed piglet cadavers in German forests (including the Hainich National Park in Thuringia), where *A. stercorosus* was counted as the most abundant species [36]. In line with these results, Jarmusz [88] detected an increase in *A. stercorosus* numbers during bloated and active decay stages of pig cadaver decomposition in forest habitats in Western Poland, peaking during advanced decay. Such late decomposition stages are characterized by substantial releases of nutrient-rich cadaveric fluids into the underlying soil [72], constituting a highly attractive food source (readily available mixture of nutrients and organic matter) for certain Geotrupidae species [36]. Collectively, these studies and our results support the importance of leaving wildlife carcasses in forest habitats to guarantee the onset and progress of later decomposition stages as an urgent precondition for preserving the diversity of important ecosystem service providers, such as the copronecrophagous dung beetles.

Particularly noticeable in our pitfall traps were the following two rare necrophilous species, the “primitive” carrion beetle *Necrophilus subterraneus* (Coleoptera: Agyrtidae) with seven individuals at fixed locations (all roe deer carcasses) and one individual at a random site (roe deer carcass) and the false clown beetle *Sphaerites glabratus* (Coleoptera: Sphaeritidae) with six specimens at random sites (five roe deer and one fox carcass) and three individuals at fixed locations (all roe deer carcasses, Table 1).

*Necrophilus subterraneus* occurs in Central Europe [89] but is not known to be in the Šumava Mountains (the Bohemian Forest, Czech Republic) so far (Růžička [90], personal communication). In the Central European mountains, *N. subterraneus* is active in summer and can be sometimes collected at carrion or other decomposing material [91]. It is nocturnal and a specialist feeder on snail carcasses (e.g., [92,93,94]). *Necrophilus* (Latreille, 1829) comprises five species in the Palearctic region [95,96]. In Europe, only *N. subterraneus* (Dahl, 1807) is present [95,97]. Such documented rare polyphagous species at ungulate carrion that is allowed to decompose entirely demonstrates the importance and necessity of regular carrion supply and retention in forest ecosystems. This can be easily achieved by exposing automatically acquired roe deer carcasses from local wildlife–vehicle collisions, as applied in the BFNP.

Furthermore, we collected nine adult individuals of the rare and only partly described (larvae are known only from the description of first instar larvae reared by Nikitsky [98]; eggs and pupae are unknown [99]) the false histerid beetle *S. glabratus* on roe deer carrion and on a red fox cadaver that was in advanced decay. Sphaeritidae are associated with decomposing organic material, such as carrion, dung, fermenting fruits, fungi, sap of dead or dying trees, and stumps [99]. In Europe, only the species *S. glabratus* is present [100,101]. For *S. glabratus*, an environmental association with northern conifer forests, sometimes at high altitudes, has been reported [99,100], which is in accordance with our forested National Park study area. All *S. glabratus* at our exposed roe deer and fox carcasses were collected along an elevational range from 774 to 880 m a.s.l. At each of 21 forest plots in an area extending from Danube lowland up to the mountain summits in the Bohemian Forest (overlapping with the same study area we used for our carrion expositions), Farwig [42] installed one trap baited with two carcasses of *Mus musculus* for attracting and collecting insect scavengers. Consistent with our results, this author trapped a total of four individuals of the uncommon species *S. glabratus* at small rodent carrion, highlighting the high conservation value of the Bohemian Forest ecosystem not only for saproxylic beetles (e.g., References [39,102]), but also for necrophilous beetles. All of these findings clearly show the importance of leaving and additionally providing wildlife carcasses—from small rodents to medium-sized carnivores up to large ungulates—in protected forests, to preserve very rare and threatened beetle species as essential members of the invertebrate part of the necrobiome.

## 5. Conclusions

Our wildlife carrion exposition and insect-trapping study clearly demonstrate the importance of concentrated carcass bait sites compared to random availability, carcass size, and decay stage on the diversity pattern of carrion-associated beetles. The exposure affected insect diversity in a different way than those reported from vertebrates. This implies the necessity of examining all parts of the necrobiome from microbes, insects, and vertebrate scavengers for a more complete understanding of the complexity in carrion visitor diversity. Furthermore, our findings of rare species on such carrion resources emphasize the importance of carrion for biodiversity conservation. Additional studies of the entire necrobiome will allow for general comparisons with communities decomposing other types of necromass, e.g., deadwood, litter, or dung, to support a more general understanding of the community structure and function of all forms of decaying organic matter.

## Figures and Tables

**Figure 1 insects-12-00412-f001:**
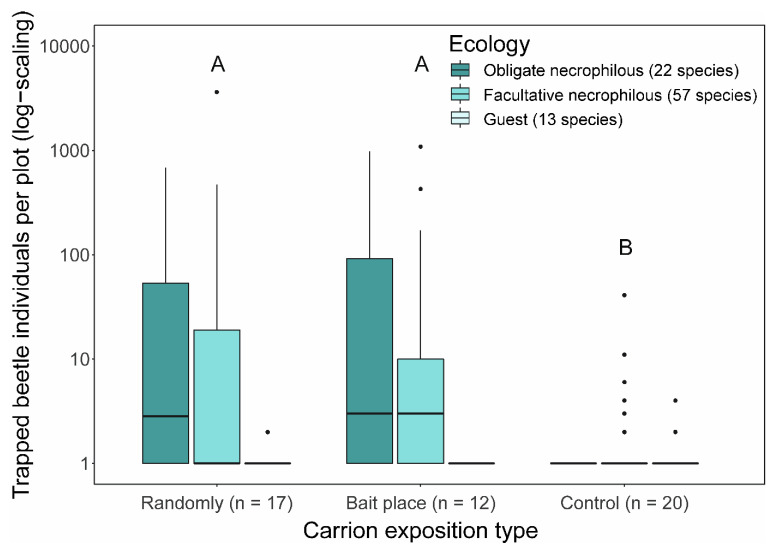
Median abundance of trapped obligate, facultative necrophilous, and guest beetle individuals per plot for each exposition type. The different letters indicate significant differences between exposition types (Poisson-GLMM with Tukey contrasts (*p* < 0.05); n = number of treatments).

**Figure 2 insects-12-00412-f002:**
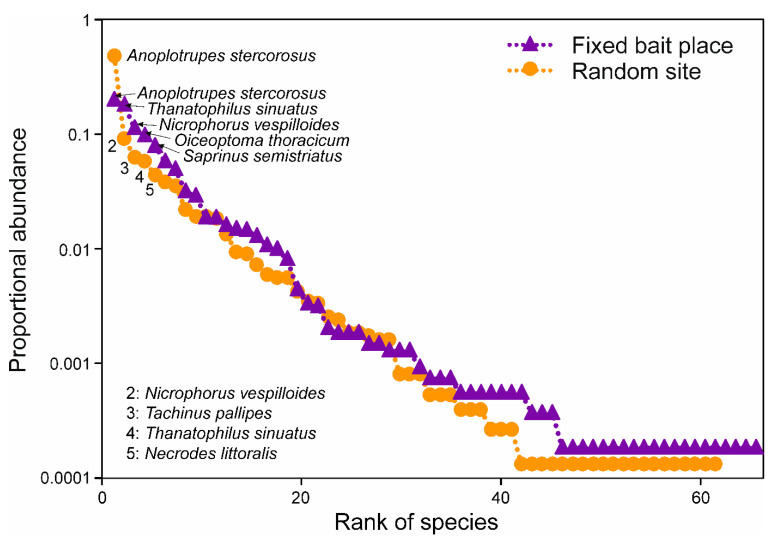
Rank–abundance curves of necrophilous beetles at random and fixed locations. Violet triangles, visitors at fixed locations; orange circles, visitors at random sites. The five most abundant species are labeled, respectively. For information regarding beetle families and -ecology (facul-tative necrophilous or obligate necrophilous) refer to Table 1.

**Figure 3 insects-12-00412-f003:**
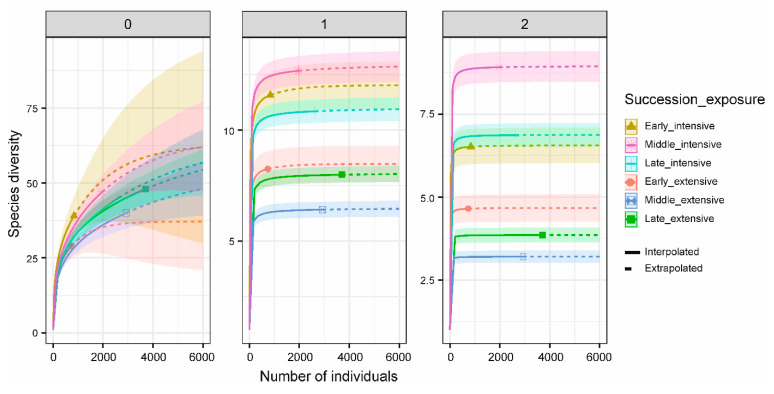
Individual-based rarefaction and extrapolation of beetle diversity for the two different treatments in combination with carcass succession stages, along with 95% unconditional confidence intervals. Species diversity was estimated for Hill numbers: q = 0 (species richness, left panel), q = 1 (exponential of Shannon’s entropy index, middle panel), and q = 2 (inverse of Simpson’s concentration index, right panel). Symbols represent the total number of reference individuals. Rarefaction curves in solid lines; extrapolation curves in dotted lines. Extensive = random site; intensive = fixed location. Early = fresh, middle = bloated and active decay, and late = advanced decay and dry remains succession stages. In transparent shading, 95% confidence intervals.

**Figure 4 insects-12-00412-f004:**
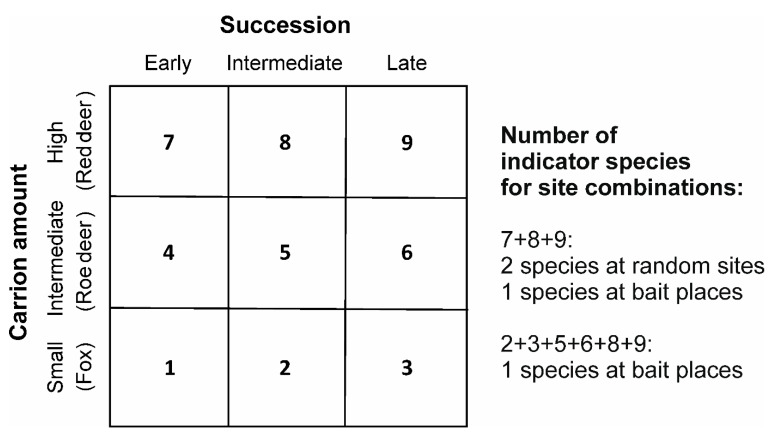
Number of indicator species for different site combinations and their associated site-group matrix used for indicator species analysis. Early = fresh decomposition stage; intermediate = bloated and active decay decomposition stages; late = advanced decay and dry remains decomposition stages.

**Table 1 insects-12-00412-t001:** Beetle species with number of individuals (Indiv) in dependency of the exposition type (random sites, fixed locations, and controls).

Species	Family	Ecology	Indiv (Random)	Indiv (Fixed)	Indiv (Controls)
*Saprinus semistriatus (L. G. Scriba, 1790)*	Histeridae	fac	165	427	3
*Margarinotus striola (C. Sahlb., 1819)*	Histeridae	fac	145	157	6
*Margarinotus brunneus (F., 1775)*	Histeridae	fac	0	1	0
*Hister unicolor L., 1758*	Histeridae	fac	1	0	0
*Sphaerites glabratus (F., 1792)*	Sphaeritidae	fac	6	3	0
*Nicrophorus humator (Gled., 1767)*	Silphidae	obl	68	58	0
*Nicrophorus investigator Zett., 1824*	Silphidae	obl	100	70	0
*Nicrophorus interruptus Steph., 1830*	Silphidae	obl	6	1	0
*Nicrophorus vespilloides Herbst, 1783*	Silphidae	obl	686	614	0
*Nicrophorus vespillo (L., 1758)*	Silphidae	obl	4	3	0
*Necrodes littoralis (L., 1758)*	Silphidae	obl	328	268	0
*Thanatophilus rugosus (L., 1758)*	Silphidae	obl	26	101	0
*Thanatophilus sinuatus (F., 1775)*	Silphidae	obl	439	976	0
*Oiceoptoma thoracicum (L., 1758)*	Silphidae	obl	290	528	0
*Necrophilus subterraneus (Dahl, 1807)*	Agyrtidae	fac	1	7	0
*Agyrtes castaneus (F., 1792)*	Agyrtidae	obl	0	0	1
*Sciodrepoides watsoni (Spence, 1813)*	Leiodidae	obl	1	0	0
*Sciodrepoides fumatus (Spence, 1813)*	Leiodidae	obl	4	0	0
*Catops kirbyi (Spence, 1813)*	Leiodidae	obl	1	1	0
*Apocatops nigrita (Er., 1837)*	Leiodidae	obl	1	0	0
*Megarthrus depressus sensu FHL Bd 4–15,*	Staphylinidae	fac	0	1	0
*Proteinus brachypterus (F., 1792)*	Staphylinidae	fac	0	3	0
*Omalium rivulare (Payk., 1789)*	Staphylinidae	fac	42	24	2
*Omalium septentrionis C. Thoms., 1857*	Staphylinidae	fac	71	87	1
*Omalium rugatum Muls. et Rey, 1880*	Staphylinidae	fac	0	1	0
*Anthobium melanocephalum (Ill., 1794)*	Staphylinidae	fac	0	7	0
*Arpedium quadrum (Grav., 1806)*	Staphylinidae	fac	14	5	4
*Oxytelus laqueatus (Marsh., 1802)*	Staphylinidae	fac	25	4	0
*Anotylus rugosus (F., 1775)*	Staphylinidae	fac	0	1	0
*Anotylus sculpturatus (Grav., 1806)*	Staphylinidae	fac	1	4	0
*Stenus clavicornis (Scop., 1763)*	Staphylinidae	guest	0	1	0
*Rugilus rufipes (Germar, 1836)*	Staphylinidae	fac	6	10	1
*Rugilus mixtus Lohse, 1956*	Staphylinidae	fac	0	1	0
*Xantholinus tricolor (F., 1787)*	Staphylinidae	fac	0	1	0
*Xantholinus longiventris Heer, 1839*	Staphylinidae	fac	3	2	1
*Atrecus affinis (Payk., 1789)*	Staphylinidae	fac	0	1	0
*Philonthus laevicollis (Lacord., 1835)*	Staphylinidae	fac	12	7	0
*Philonthus rufipes (Steph., 1832)*	Staphylinidae	fac	13	11	0
*Philonthus carbonarius Grav., 1802*	Staphylinidae	fac	266	102	0
*Philonthus addendus Sharp, 1867*	Staphylinidae	fac	1	0	0
*Philonthus pseudovarians A. Strand, 1941*	Staphylinidae	fac	0	4	0
*Philonthus splendens (F., 1792)*	Staphylinidae	fac	3	0	1
*Philonthus fimetarius (Grav., 1802)*	Staphylinidae	fac	143	79	1
*Philonthus marginatus (O. F. Müller, 1764)*	Staphylinidae	fac	18	10	0
*Creophilus maxillosus (L., 1758)*	Staphylinidae	fac	137	172	0
*Ontholestes tessellatus (Geoffr., 1785)*	Staphylinidae	fac	19	17	0
*Ontholestes murinus (L., 1758)*	Staphylinidae	obl	0	3	0
*Dinothenarus fossor (Scop., 1771)*	Staphylinidae	guest	0	0	4
*Ocypus olens (O. F. Müller, 1764)*	Staphylinidae	fac	0	0	1
*Quedius cinctus (Payk., 1790)*	Staphylinidae	fac	55	18	0
*Lordithon trinotatus (Er., 1839)*	Staphylinidae	fac	32	3	11
*Tachyporus pusillus Grav., 1806*	Staphylinidae	fac	1	0	0
*Tachinus pallipes Grav., 1806*	Staphylinidae	fac	471	81	2
*Atheta excellens (Kr., 1856)*	Staphylinidae	fac	1	0	0
*Atheta longicornis (Grav., 1802)*	Staphylinidae	fac	0	1	0
*Acrotona parvula (Mannerh., 1830)*	Staphylinidae	fac	1	0	0
*Oxypoda opaca (Grav., 1802)*	Staphylinidae	fac	0	1	0
*Oxypoda formosa Kr., 1856*	Staphylinidae	fac	1	0	0
*Aleochara curtula (Goeze, 1777)*	Staphylinidae	obl	1	2	0
*Lampyris noctiluca (L., 1767)*	Lampyridae	fac	1	0	0
*Necrobia violacea (L., 1758)*	Cleridae	obl	1	44	0
*Necrobia rufipes (DeGeer, 1775)*	Cleridae	obl	2	3	0
*Dermestes murinus L., 1758*	Dermestidae	obl	2	1	0
*Dermestes laniarius Ill., 1801*	Dermestidae	obl	0	1	0
*Byrrhus pilula (L., 1758)*	Byrrhidae	guest	0	0	4
*Byrrhus glabratus Heer, 1841*	Byrrhidae	guest	0	0	1
*Omosita depressa (L., 1758)*	Nitidulidae	obl	14	314	0
*Cychramus variegatus (Herbst, 1792)*	Nitidulidae	guest	2	0	0
*Mycetina cruciata (Schaller, 1783)*	Endomychidae	guest	1	0	0
*Anaspis rufilabris (Gyll., 1827)*	Scraptiidae	fac	1	0	0
*Corticeus unicolor Pill. et Mitt., 1783*	Tenebrionidae	guest	0	0	1
*Trox scaber (L., 1767)*	Trogidae	obl	0	1	0
*Geotrupes stercorarius (L., 1758)*	Geotrupidae	fac	0	1	0
*Anoplotrupes stercorosus (Hartmann in L. G. Scriba, 1791)*	Geotrupidae	fac	3625	1086	41
*Teuchestes fossor (L., 1758)*	Scarabaeidae	fac	0	1	0
*Ammoecius brevis (Er., 1848)*	Scarabaeidae	fac	4	8	0
*Acrossus rufipes (L., 1758)*	Scarabaeidae	fac	45	54	0
*Acrossus luridus (F., 1775)*	Scarabaeidae	fac	1	0	0
*Acrossus depressus (Kugel., 1792)*	Scarabaeidae	fac	42	10	0
*Limarus maculatus (Sturm, 1800)*	Scarabaeidae	fac	12	3	0
*Volinus sticticus (Panzer, 1798)*	Scarabaeidae	fac	0	1	0
*Nimbus contaminatus (Herbst, 1783)*	Scarabaeidae	fac	3	0	0
*Melinopterus sphacelatus (Panzer, 1798)*	Scarabaeidae	fac	0	2	0
*Melinopterus prodromus (Brahm, 1790)*	Scarabaeidae	fac	1	8	0
*Planolinus fasciatus (Olivier, 1789)*	Scarabaeidae	fac	1	0	0
*Calamosternus granarius (L., 1767)*	Scarabaeidae	fac	0	1	0
*Serica brunnea (L., 1758)*	Scarabaeidae	guest	0	0	1
*Ips typographus (L., 1758)*	Scolytidae	guest	0	0	1
*Otiorhynchus scaber (L., 1758)*	Curculionidae	guest	1	0	0
*Phyllobius arborator (Herbst, 1797)*	Curculionidae	guest	0	1	1
*Hylobius abietis (L., 1758)*	Curculionidae	guest	0	0	2
*Plinthus tischeri Germar, 1824*	Curculionidae	guest	0	0	1

In Ecology, “obl” stands for obligatory necrophilous, “fac” stands for facultative necrophilous, and “guest” designates random visitors (non-necrophilous species) at carrion exposition sites.

## Data Availability

The data presented in this study are available on request from the corresponding author or from the Bavarian Forest National Park Administration.

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
