# Peer review of "Carcasses at Fixed Locations Host a Higher Diversity of Necrophilous Beetles"

_insects, 2021, doi:10.3390/insects12050412_

Round 1

Reviewer 1 Report

a.

beautifully done 

b.

somewhere after line 120, pls add reference:

http://www.wiki2.benecke.com/images/5/5f/Fiedler_Halbach_Sinclair_Benecke_What_is_the_edge_of_a_forest.pdf

c.

lines 160 ff:

italics for latin species names, pls (acer, cervus, lynx, vulpes...)

d.

text to figs 1 & 2 & 4 & line 486:

is "obligate-", "necrophilous-", "random-", "decay-", "exposition-" correct in english? might be a translation bug?

e.

lines 275 & 276; 306-308 ff.:

pls check if "," and "." are used properly in numbers given there to avoid european / american system troubles

f.

line 395, 415:

Staphilinid (with capital "S")? "Scarabaeidea" is written with capital "S"; maybe this should be a decision by editors? (i am not sure.)

g.

line 497:

replace "xxx"?

Reviewer 2 Report

This is an excellent study adressing a new & understudied field of research (carrion ecology and more specifically the effect of carrion gathering on local insectes biodiversity).

M&M are well designed and results are clear and properly reported.

The discussion is interestings and provides a lot of usefull informations on the species found. The authors also suggest how to manage roadkill carcasses to increase biodiversity.

Minor comments are reported in the enclosed annotated manuscript.

Reviewer 3 Report

This paper is well written, and will serve as an important manuscript for those researchers designing similar studies. 
